# How do we tread? Differences in stability-related foot placement control between overground and treadmill walking in young adults

Charlotte Lang[1], Deepak K. Ravi [1], Sjoerd M. Bruijn[2,3,4], Jeffrey M. Hausdorff[5,6], Jaap H. van Dieën [2,4], Anina Moira van Leeuwen [2,3,4]*

1 Laboratory for Movement Biomechanics, Institute for Biomechanics, Department of Health Sciences& Technology, ETH Zürich, Zürich, Switzerland, 2 Department of Human Movement Science, Vrije Universiteit Amsterdam, Faculty of Behavioural and Movement Sciences, Amsterdam, Netherlands, 3 Institute of Brain and Behavior, Amsterdam, Netherlands, 4 Amsterdam Movement Sciences, Research Program, Amsterdam, Netherlands, 5 Center for the Study of Movement, Cognition, and Mobility, Neurological Institute, Tel Aviv Medical Center, Tel Aviv, Israel, 6 Department of Physical Therapy, Faculty of Medical & Health Sciences, Tel Aviv University, Tel Aviv, Israel

* moiravleeuwen@gmail.com

## Abstract

Step-by-step foot placement control, accommodating for natural variations in center-of-mass state, ensures stability during steady-state gait. Current understanding of this foot placement mechanisms is primarily based on findings during treadmill walking. However, contextual differences might hamper generalization of these findings towards overground walking, and ultimately daily life gait. This study investigated whether foot placement control manifests itself differently during overground as compared to treadmill walking in healthy young adults. 14 young adults walked at comfortable walking speed, both on the treadmill and overground in a figure-8 path. During overground walking we found a significant relationship between the step width/step length and the center-of-mass state during the swing phase of walking, capturing foot placement control with the same linear model as during treadmill walking. Contrary to what was hypothesized, we found a significant lower foot placement precision during overground walking for the step width model with center-of-mass state at the start of the swing phase as predictor, complemented by a wider average step width. Moreover, during overground walking participants responded less strongly to a deviation in center-of-mass position, yet significantly stronger to deviations in center-of-mass velocity at the end of the step for both the step width/step length models. Exploratory analysis showed a larger relative contribution of velocity feedback during overground walking as compared to treadmill walking. These differences warrant caution in generalizing foot placement findings during treadmill walking to overground walking and might be promising for the estimation of foot placement control in daily life gait.

**Data availability statement:** At this point in time the data cannot be shared publicly because the data was not collected by the co-authors. As such we do not have permission to share the data. The researchers who allowed us to use their data are still working on their own data analysis. The data can be made available upon request by contacting Prof. dr. William R. Taylor (bt@ethz.ch).

**Funding:** This research was financially supported by the EU Joint Programme – Neurodegenerative Disease Research (JPND) to the StepuP consortium: Steps against the burden of Parkinson's Disease,grant number JPND2022-128, obtained by the StepuP consortium (J.H. van Dieën, W. Maetzler, J. Hausdorff, M. Brodie, N. Singh & F. Laporta). The local funding agencies awarding the funding for this JPND project to authors of this paper were ZonMw, Swiss National Science Foundation and Israel Ministry of Health. Moreover, the experimenters responsible for the data collection were funded by the LOOP Zurich and the Vontobel Stiftung. The funders had no role in study design, data collection and analysis, decision to publish, or preparation of the manuscript.

**Competing interests:** The authors have declared that no competing interests exist.

## 1. Introduction

During human walking, stability is maintained through effective coordination between the center-of-mass (CoM) and its base of support. To overcome the challenge of a highly positioned CoM above a small base of support, neural control mechanisms ensure a stable gait [1]. The most dominant mechanism for maintaining stability during steady-state walking is foot placement control [2]. By adjusting step width and step length to accommodate variations in the CoM state (i.e., its position and velocity) [3–5], the foot placement mechanism helps maintain gait stability. However, with old age, or in certain patient populations, foot placement control can be impaired [3,6–8], which may contribute to an increased risk of falls [6]. To be able to put such impairments into context, foot placement control has been extensively studied in healthy adults [9–12], to firstly gain an understanding of the unimpaired mechanism. Yet, current insights into this mechanism predominantly come from treadmill-based studies. Understanding how foot placement control differs under various measurement conditions, particularly those that more closely resemble daily-life walking, is a critical step towards understanding real-world stability control. To help gain such understanding this study focuses on foot placement control mechanisms in healthy young adults, during both overground and treadmill walking.

Step-by-step foot placement control has been characterized using a linear model that correlates foot placement with the CoM state during the preceding swing phase, in both the mediolateral (ML) and anteroposterior (AP) directions [9,13]. This foot placement model shows that, during the swing phase, the CoM state predicts subsequent foot placement, and, during early swing, does so even better than the kinematic state of the swing foot itself [9]. Combined with evidence that sensory perturbations affect foot placement [3,14], this suggests that the linear foot placement model represents a feedback control mechanism in which sensory information on CoM state serves as input [4]. The model can be used to assess the degree of foot placement control. The gain of the model can be considered a measure of the strength of foot placement responses to CoM deviations and the residual of the model as a measure of foot placement precision, i.e., how precise the feedback mechanism is executed. These measures have shown to capture foot placement training effects during treadmill walking [15,16]. However, it is less clear how these control characteristics manifest under different walking conditions, such as during overground walking.

The foot placement model described above captures step-by-step adjustments in step width and length to accommodate variations in CoM state during steady-state walking. However, other foot placement mechanisms contribute to maintaining walking stability as well [2]. Stepping strategies, such as controlling average step width/length and stride frequency, also play a role in maintaining gait stability [17,18]. For example, walking with a wider average step width reduces the need for tight step-by-step foot placement control to achieve comparable gait stability [19]. Moreover, mechanisms other than foot placement control, such as ankle moment and angular momentum control, can compensate for foot placement errors [20], or be recruited when stability demands increase [21]. Given these complementary stability mechanisms, the expression of step-by-step foot placement control is likely

context-dependent. Treadmill walking presents different task constraints than overground walking, especially when the treadmill operates at a constant speed. The fixed belt-speed limits variations in step length. Moreover, treadmill walking lacks optical flow and imposes greater stepping constraints: foot placement is constrained to the treadmill surface. These constraints may result in different stabilizing strategies and ultimately alter how foot placement control manifests. Differences in foot placement control have already been reported between walking on a split-belt treadmill (with belts tied to the same speed) and walking on a regular single-belt treadmill [22], with wider steps and less tight step-by-step foot placement control on a split-belt treadmill. Given the evidence that even subtle changes in walking context can alter foot placement control, differences between overground and treadmill walking can be expected.

First, we hypothesize that foot placement is controlled in relation to the CoM state on a step-by-step basis during overground walking, analogous to treadmill walking (H1). Second, we hypothesize differences in precision and strength of step-by-step foot placement control between overground and treadmill walking. We hypothesize a higher precision of foot placement control (i.e., smaller foot placement errors, H2) and stronger foot placement responses (i.e., a larger adjustment in foot placement for a given deviation in CoM state, H3) during overground walking as compared to treadmill walking, given that there are fewer stepping constraints competing with stability-related foot placement control during overground walking.

As mentioned above, step-by-step foot placement control is modulated by changes in average step width and similarly by changes in gait speed [23]. Therefore, we compared foot placement control between treadmill and overground walking while accounting for differences in speed and step width, to ensure that any observed effects reflected the walking condition itself rather than these gait characteristics.

## 2. Methods

This study used a previously collected dataset from the Laboratory for Movement Biomechanics at ETH Zürich, approved by the relevant ethics committee (Approval no: 2022–01382), with all participants providing written informed consent. The data was made accessible for the authors of this manuscript on 03-03-2025, devoid of information that could identify individual participants. The dataset comprised 14 young, healthy participants (8 males; mean age: 25.1±2.5 years). As part of the study, participants walked on a treadmill for six minutes and completed an overground figure-8 walking task around cones positioned 10 meters apart for a continuous period of four minutes. Both walking tasks were performed at comfortable, self-selected speed. Kinematic data were recorded at a sampling rate of 200 Hz for treadmill walking and 100 Hz for overground walking using a three-dimensional motion capture system (consisting of 14 cameras; 61 markers; Vicon Nexus, versions 2.3/2.8.2, Oxford Metrics, United Kingdom). For the overground figure-8 walking task, only the straight-path segments were recorded and analyzed. The last step of each straight path segment was removed and an array containing all valid steps was then generated for the analysis.

To keep walking as natural as possible, no specific speed or stepping constraints were imposed, neither during treadmill nor overground walking. For the treadmill trial, the belt speed was set to each participant's comfortable pace by gradually increasing the speed. The final speed was calculated as the average of the speeds at which the participant reported feeling slightly too fast and slightly too slow. For the overground trial, participants were instructed to walk at their comfortable speed, which did not necessarily match their comfortable treadmill walking speed.

### 2.1. Data extraction/preparation

#### 2.1.1. Rotation of data. To express the overground walking paths in a local coordinate system aligned with the walking direction, marker trajectories recorded in the laboratory coordinate system were rotated. The walking direction was determined by calculating the average angle of the pelvis in the horizontal plane during each straight path segment, i.e., the average orientation of the pelvis relative to the global coordinate system (Fig 1). This angle was then used to construct a rotation matrix, which was applied to transform the marker trajectories from the laboratory coordinate system into the newly aligned coordinate system.

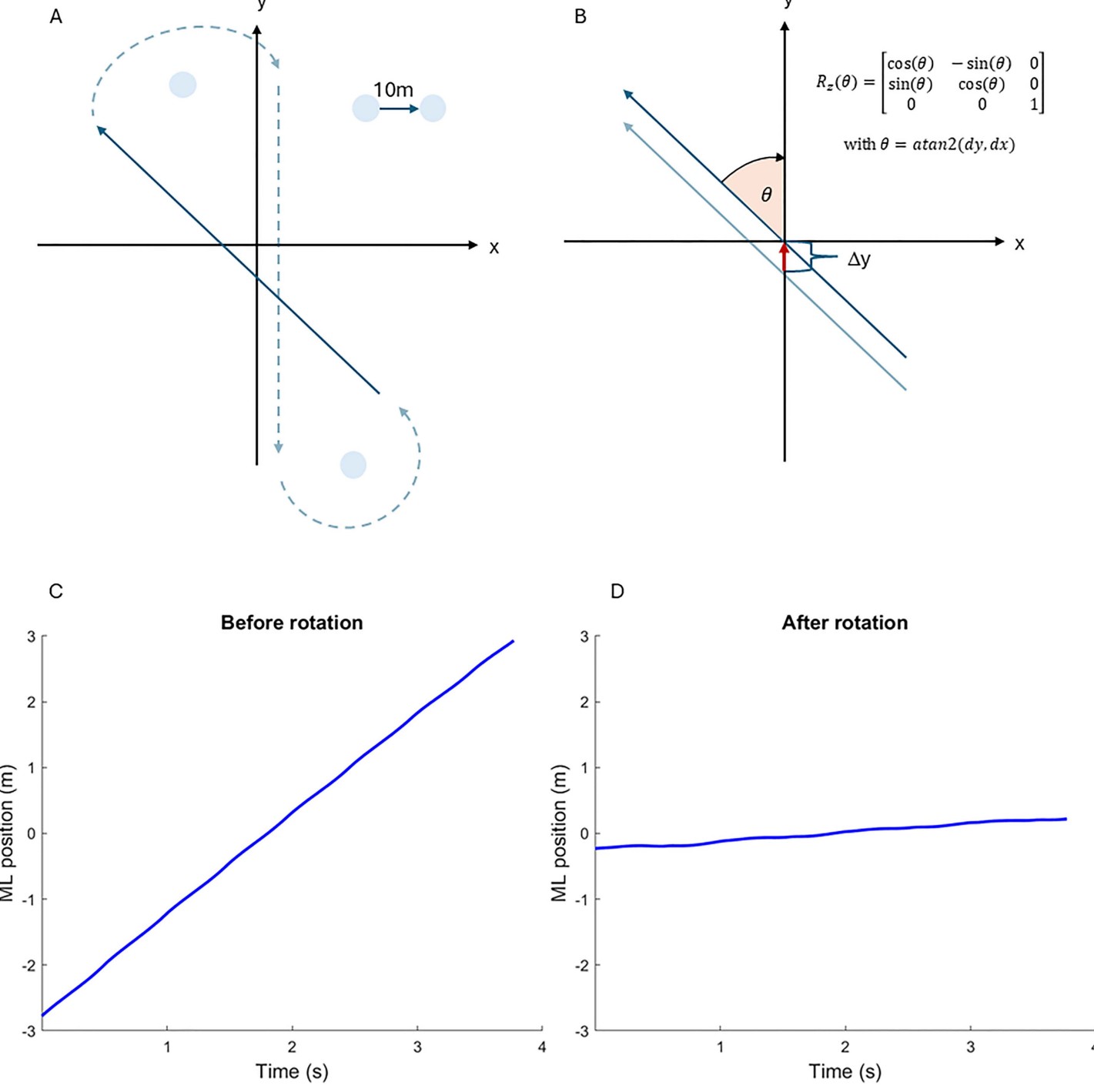

**Fig 1. Illustration of the figure-8 walking path and construction of a coordinate system along the straight part of the walking path.** The figure-8 walking path in the global coordinate system is illustrated in panel A. The walking path angle was defined over each trial including 8-10 steps with respect to the global coordinate system. The formula for computing the rotation matrix based on this angle is presented in panel B. Panel C shows the ML walking path before rotation, panel D after rotation.

**2.1.2. Step definitions.** Gait events (heel strike and toe-off) were detected using a custom algorithm based on foot velocity. CoM and foot trajectories were extracted from marker data, with heel markers used for foot trajectories. The CoM was defined according to the best available data quality, using either the sacral marker, the mean of left and right posterior superior iliac markers, or the mean of left and right anterior superior iliac markers. The same definition was applied consistently across all trials of each individual subject. Swing phases (toe-off to heel strike) were identified based on the detected gait events and subsequently time-normalized to 0–100% (51 samples) using spline interpolation to ensure comparability across steps of varying time durations. This process was applied to all three movement directions. The last step of each trial was excluded, and all walking trials were combined afterwards in an array containing all valid steps. For all participants and both conditions (treadmill and overground), the same number of steps (n = 130) was included in the analysis.

## 2.2. Foot placement models

Step-by-step foot placement control was quantified using outcome measures derived from a foot placement model [1], (equations 1,2). This model predicts either step width (SW; equation 1) or step length (SL; equation 2) based on the CoM state for each step defined by ML/AP CoM position ($CoM_{pos\_ML}$, $CoM_{pos\_AP}$) and ML/AP CoM velocity ($CoM_{vel\_ML}$, $CoM_{vel\_AP}$). Step width was defined as the ML distance between the two heel markers of the feet, where the position of each foot was determined at the instance of their respective midstance, in order to ensure that the foot was flat on the ground. Step length was defined in a similar way but in the AP direction, and the positions of both feet were determined at the instance of heel strike of the leading foot, to avoid treadmill movement altering the AP distance. $CoM_{pos}$ was defined as the ML or AP position of the pelvis marker/superior iliac markers with respect to stance foot for each time-normalized sample (i = 1:51) of the swing phase, and $CoM_{vel}$ as its derivative. All variables were demeaned prior to fitting the following linear models.

$$SW = \beta_{pos_{ML}} * CoM_{pos_{ML}}(i) + \beta_{vel_{ML}} * CoM_{vel_{ML}}(i) + \varepsilon_{ML} \tag{1}$$

$$SL = \beta_{pos_{AP}} * CoM_{pos_{AP}}(i) + \beta_{vel_{AP}} * CoM_{vel_{AP}}(i) + \varepsilon_{AP} \tag{2}$$

Where $\beta_{posML}$, $\beta_{velML}$, $\beta_{posAPL}$ and $\beta_{velAP}$ are the regression coefficients (or gains) of the model and denote the "strength of the foot placement responses" to deviations in CoM state. $\varepsilon_{ML}$ and $\varepsilon_{AP}$ are the residuals of the respective models and their standard deviation will be referred to as "foot placement errors" or the precision of foot placement control. The magnitude of these foot placement errors can be computed by taking the standard deviation, and this will be considered as a measure of the precision of foot placement control. Each sample (i) of the swing phase corresponds to a different normalized time point of the swing phase, with the last sample (i = 51) coinciding with heel strike (i.e., the instance of foot placement). The outcomes of the model for i = 51 can be interpreted as performance measures of stability-related foot placement control. Samples earlier in the swing phase reflect the extent to which swing phase kinematic state can predict upcoming foot placement, and, models fitted at these points can be interpreted as measures of feedback control. To capture both feedback control and stability-related foot placement performance, here, we considered outcome measures (i.e., the gains, the standard deviation of the residuals and the partial correlations) derived from ML and AP foot placement models at the start of the step (i = 1) and at heel strike (i = 51).

Measures derived from the model fitted for i = 1 (i.e., with the CoM state at the start of the step as predictors) can be considered as measures of feedback control, since at the start of the step the CoM state has been shown to be a better predictor of imminent foot placement than the state of the swing foot itself [9]. This suggests early feedback control based on sensory information of the CoM state. Moreover, muscle activity during early swing drives step-by-step foot placement control [10,12], providing further support of feedback control relying on early swing CoM state information. At heel strike

(i = 51), i.e., the instance of foot placement, the model captures the result of this feedback control. At the instance of foot placement, the coordination between the CoM and its base-of-support is realized [2,9,24]. Therefore, measures derived from the model fitted for i = 51 (i.e., with the CoM state at heel strike) can be considered as performance measures of stability-related foot placement control.

## 2.3. Statistics

All statistical analyses were conducted independently for SW (mediolateral, equation 1) and SL (anteroposterior) (equations 1&2), and separately for the start of the step (i = 1) and at heel strike (i = 51). We considered an alpha level of 0.05 for significance. All analyses were done in Matlab (version R2023b, The MathWorks Inc., Natick) and JASP (JASP 0.18.3).

**2.3.1. Primary statistics.** To test our first hypothesis that foot placement is controlled in relation to the CoM state on a step-by-step basis during overground walking (H1), we tested whether the foot placement models (equations 1&2) demonstrated a significant relationship by testing all regression coefficients ($\beta_{posML}$, $\beta_{velML}$, $\beta_{posAP}$, $\beta_{posAP}$) against 0 in one-sample t-tests on group level.

Paired t-tests were used to test whether gait speed, step width and step length differed between overground and treadmill walking, prior to testing H2 and H3. Significant differences in these variables between treadmill and overground walking identified potential cofounders which have to be accounted for in the linear mixed models used to test H2 and H3.

To test our second hypothesis that foot placement precision would be higher during overground as compared to treadmill walking (H2), we used a linear mixed model with *Foot Placement Error Magnitude* as the dependent variable, *Gait type ("treadmill","overground")* as Fixed effect variable, and a random intercept for participants. Since we did not control walking speed and step width between treadmill and overground walking conditions, and these gait parameters can influence the degree of step-by-step foot placement control [18,21], we also included Δ *Step Width*/Δ *Step Length* (*StepWidth/StepLength_overground – StepWidth/StepLength_treadmill*), Δ *Speed* (*Speed_overground – Speed_treadmill*) and the interaction Δ *Step Width*\*Δ *Speed* as Fixed effect variables to control for potential step width and speed effects.

To test our third hypothesis that foot placement responses would be stronger during overground as compared to treadmill walking (H3), we used a linear mixed models with respectively $\beta_{pos}$ and $\beta_{vel}$ as dependent variables and *Gait type ("treadmill","overground")*, Δ *Step Width*/Δ *Step Length*, Δ *Speed* and Δ *Step Width*/Δ *Step Length*\*Δ *Speed* as Fixed effect variables.

**2.3.2. Exploratory analysis.** Given the results of comparing the gains ($\beta_{posML}$, $\beta_{velML}$, $\beta_{posAP}$, $\beta_{posAP}$) between treadmill and overground walking, we decided to also compute partial correlations for $CoM_{pos}$ and $CoM_{vel}$, as a measure of the relative contribution of position and velocity feedback to foot placement control. We used a linear mixed models with $\rho_{CoM\_pos}$ and $\rho_{CoM\_vel}$ as dependent variables and *Gait type ("treadmill","overground")*, Δ *Step Width*/Δ *Step Length*, Δ *Speed* and Δ *Step Width*/Δ *Step Length*\*Δ *Speed* as Fixed effect variables.

## 3. Results

The data of 13 out of 14 participants were successfully preprocessed. One participant was identified as an outlier and was therefore excluded from all further analyses. During overground walking, the foot placement error for this participant was approximately seven times higher than that of the other participants, suggesting that either there had been a measurement error, or the behavior of this participant was so different that it was not comparable to the other participants.

### 3.1. Primary outcome measures

**3.1.1. Foot placement control during overground walking.** During overground walking, we found significant positive values for both $\beta_{pos}$ and $\beta_{vel}$ at the start of the step (ML: *p < 0.001; p < 0.001,* AP: *p = 0.04; p < 0.001*) and at heel strike (ML: *p < 0.001; p < 0.001,* AP: *p = 0.022; p < 0.001*) in both the ML and AP directions (Fig 2). This indicates a significant relationship of step width with both $CoM_{pos\_ML}$ and $CoM_{vel\_ML}$ as well as of step length with both $CoM_{pos\_AP}$ and

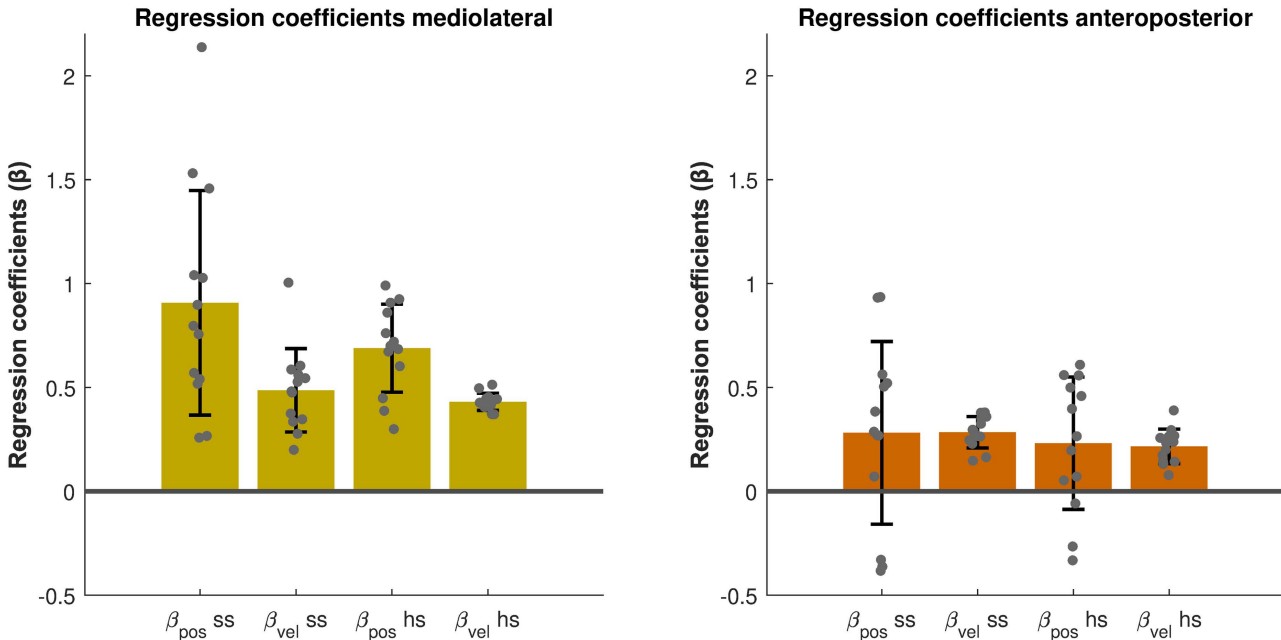

**Fig 2. Regression coefficients ($\beta_{pos}$, $\beta_{vel}$) of the ML and AP foot placement models during overground walking.** Mean regression coefficients are depicted for the predictors at the start of the step and at heel strike. Error bars represent the standard deviation and grey dots are the individual data points.

$CoM_{vel\_AP}$. Specifically, more (faster) lateral/forward deviations of the CoM are followed by wider/longer steps, whereas more (slower) medial/backward deviations of the CoM are followed by narrower/shorter steps.

**3.1.2. Potential cofounders.** Walking speed was significantly higher during overground walking compared with treadmill walking *(p < 0.001)* (Fig 3, left). Step width and step length were also significantly greater during overground walking *(both p < 0.001*; Fig 3, middle and right).

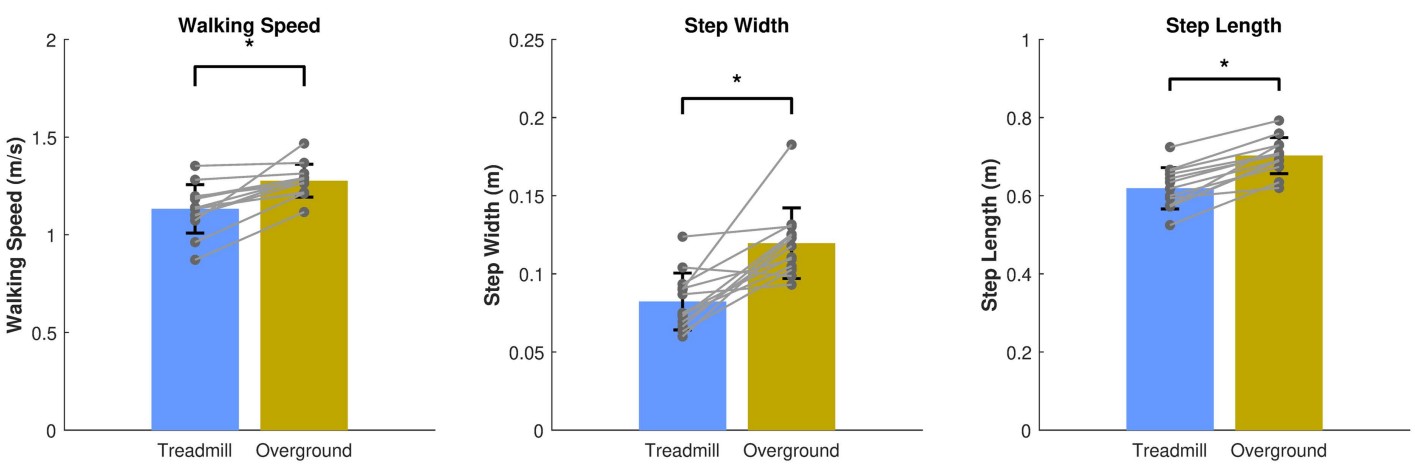

**Fig 3. Walking speed, Step width and Step length during treadmill and overground walking. Mean walking speed (m/s), step width (m) and step length (m) are depicted.** Error bars represent the standard deviation and grey dots the individual data points.*p < 0.05.

**3.1.3. Differences in precision of foot placement control between treadmill and overground walking.** In the ML direction, after adjusting for gait speed and step width, we found a significant effect of Gait type ("treadmill", "overground") at the start of the step ($p < 0.05$), indicating significantly larger foot placement errors during overground walking as compared to treadmill walking (Fig 4). However, at the end of the step and at both gait phases in the AP direction, no significant effect of Gait type was observed.

**3.1.4. Differences in the strength of foot placement responses between treadmill and overground walking.** For $\beta_{pos}$ and $\beta_{vel}$, we found no significant effect of Gait type at the start of the step in either movement directions after adjusting for gait speed and step width (Fig 5). Only $\beta_{vel}$ in the AP direction was significantly affected by Gait Speed ($p < 0.05$). However, at heel strike, there was a significant main effect of Gait type for both $\beta_{pos}$ and $\beta_{vel}$. Specifically, $\beta_{pos}$ was significantly smaller during overground as compared to treadmill walking in both the ML ($p < 0.001$) and AP ($p < 0.05$) directions. In addition, the ANOVA revealed a significant interaction of Gait Speed ($p < 0.05$) and Step Length*Gait Speed ($p < 0.05$) at heel strike in the AP direction. Conversely, we observed an opposite trend for $\beta_{vel}$ at the end of the step, with significantly higher values during overground compared to treadmill walking ($p < 0.001$, $p < 0.05$).

## 3.2. Exploratory analyses

**3.2.1. Contribution of position and velocity feedback.** $\beta_{pos}$ was lower when walking overground compared to walking on a treadmill (see Fig 5), whereas $\beta_{vel}$ showed an increase. Given these opposite changes, we explored whether the relative contributions of $CoM_{pos}$ and $CoM_{vel}$ in predicting foot placement also differed, by assessing differences in the partial correlations of $CoM_{pos}$ and $CoM_{vel}$ with step width/length.

In line with $\beta_{pos}$, we found no effect of Gait type at the start of the step for $CoM_{pos}$ in both ML and AP directions (Fig 6). However, at heel strike there was a significant main effect of Gait type for the partial correlations of both $CoM_{pos}$ and $CoM_{vel}$. Specifically, the partial correlation of $CoM_{pos}$ was lower during overground as compared to treadmill walking ($p < 0.005$, $p < 0.05$). In ML direction, there was a significant Δ *Step Width* *Δ *Speed* interaction ($p < 0.05$) and in AP direction a significant interaction of Gait Speed ($p < 0.05$) was observed. In contrast, the partial correlation of $CoM_{vel}$ was higher during

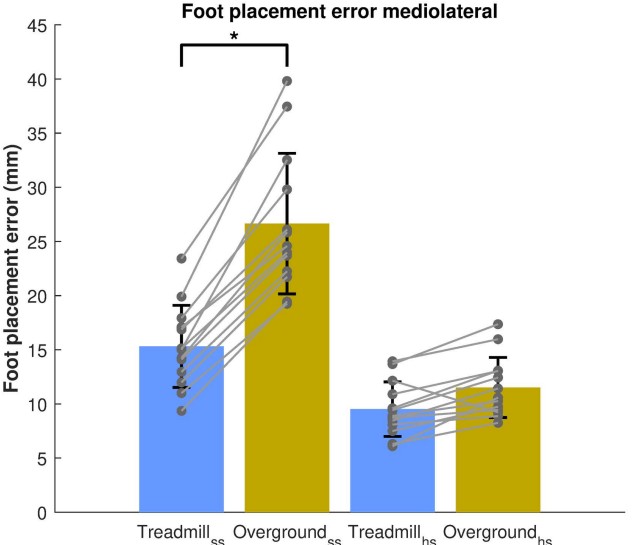
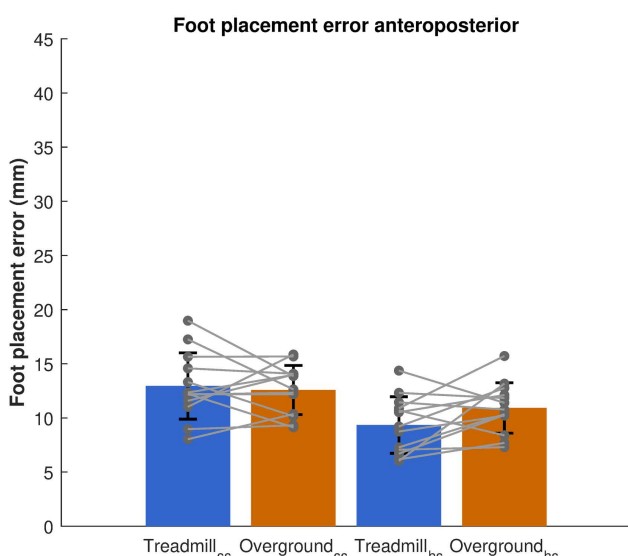

**Fig 4. Comparison of ML and AP foot placement errors between treadmill and overground walking.** Foot placement errors (i.e., the standard deviation of the residuals in millimeters), as a measure for foot placement precision are depicted for the predictions at the start of the step and at heel strike. Error bars represent the standard deviation and grey dots the individual data points. *$p < 0.05$.

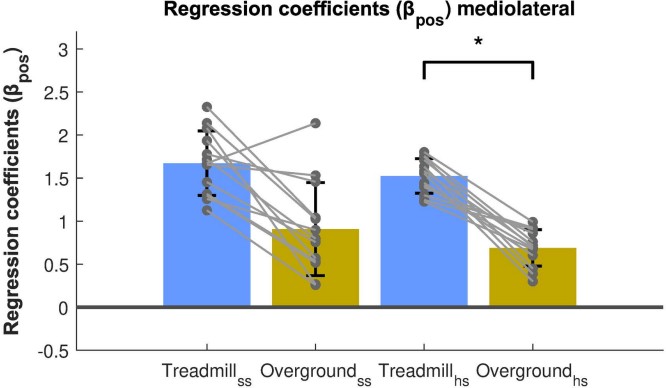
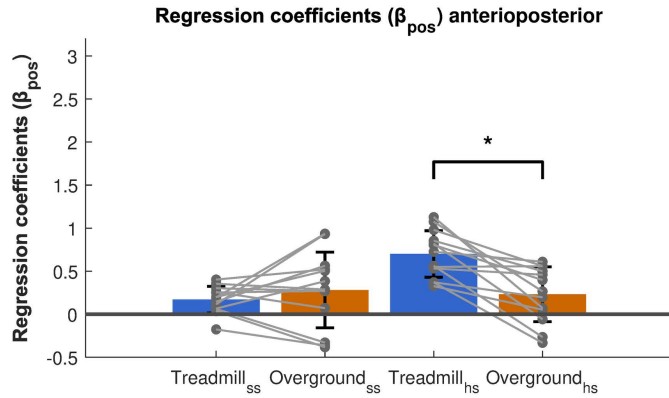
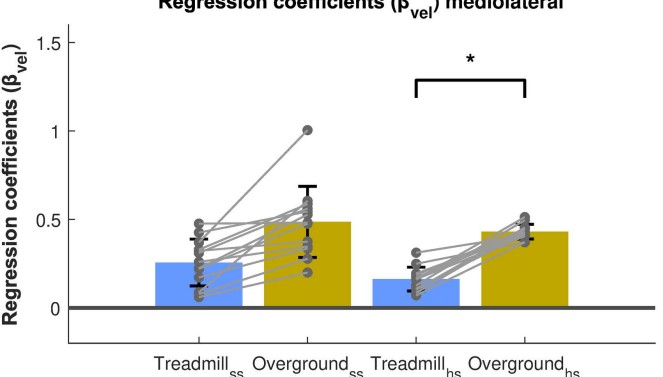
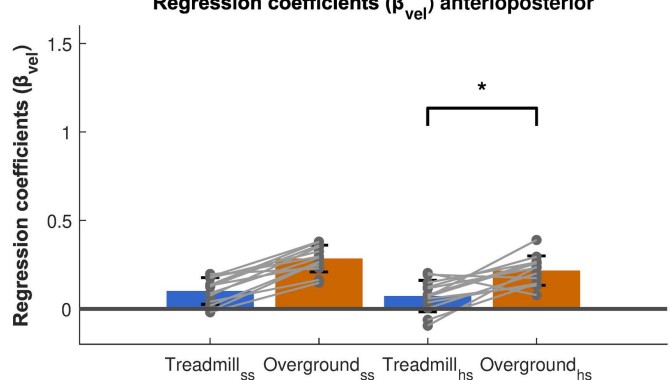

**Fig 5. Regression coefficients of CoM$_{pos}$ and CoM$_{vel}$ of the mediolateral and anteroposterior foot placement models, comparing treadmill and overground walking.** Mean regression coefficients are depicted for the predictions at the start of the step and at heel strike. Error bars represent the standard deviation and grey dots the individual data points. *$p < 0.05$.

overground as compared to treadmill walking at the start of the step in ML direction ($p < 0.05$) and at heel strike ($p < 0.001$, $p < 0.01$).

During treadmill walking, for both the start of the step and at heel strike, the partial correlation of CoM$_{vel}$ was lower compared to the partial correlation of CoM$_{pos}$ ($p < 0.001$; $p < 0.001$). Conversely during overground walking, for both the start of the step and at heel strike, the partial correlation of CoM$_{vel}$ was higher compared to the partial correlation of CoM$_{pos}$ ($p = 0.002$; $p < 0.001$).

## 4. Discussion

One of the stability mechanisms that has been extensively studied during treadmill walking is foot placement control [5,10,14]. In both the ML and AP directions, we adjust our foot placement on every step to accommodate variations in the CoM state, thereby maintaining gait stability. In this study, we tested to what extent this important stability mechanism is utilized during overground compared to treadmill walking. We found a significant relationship between step width and both CoM$_{pos\_ML}$ and CoM$_{vel\_ML}$, as well as between step length and both CoM$_{pos\_AP}$ and CoM$_{vel\_AP}$. These results confirm that step-by-step foot placement control is used during overground walking. However, the mechanism manifests differently in overground compared to treadmill walking. We found significantly lower ML foot placement precision and a larger velocity gain during overground compared to treadmill walking.

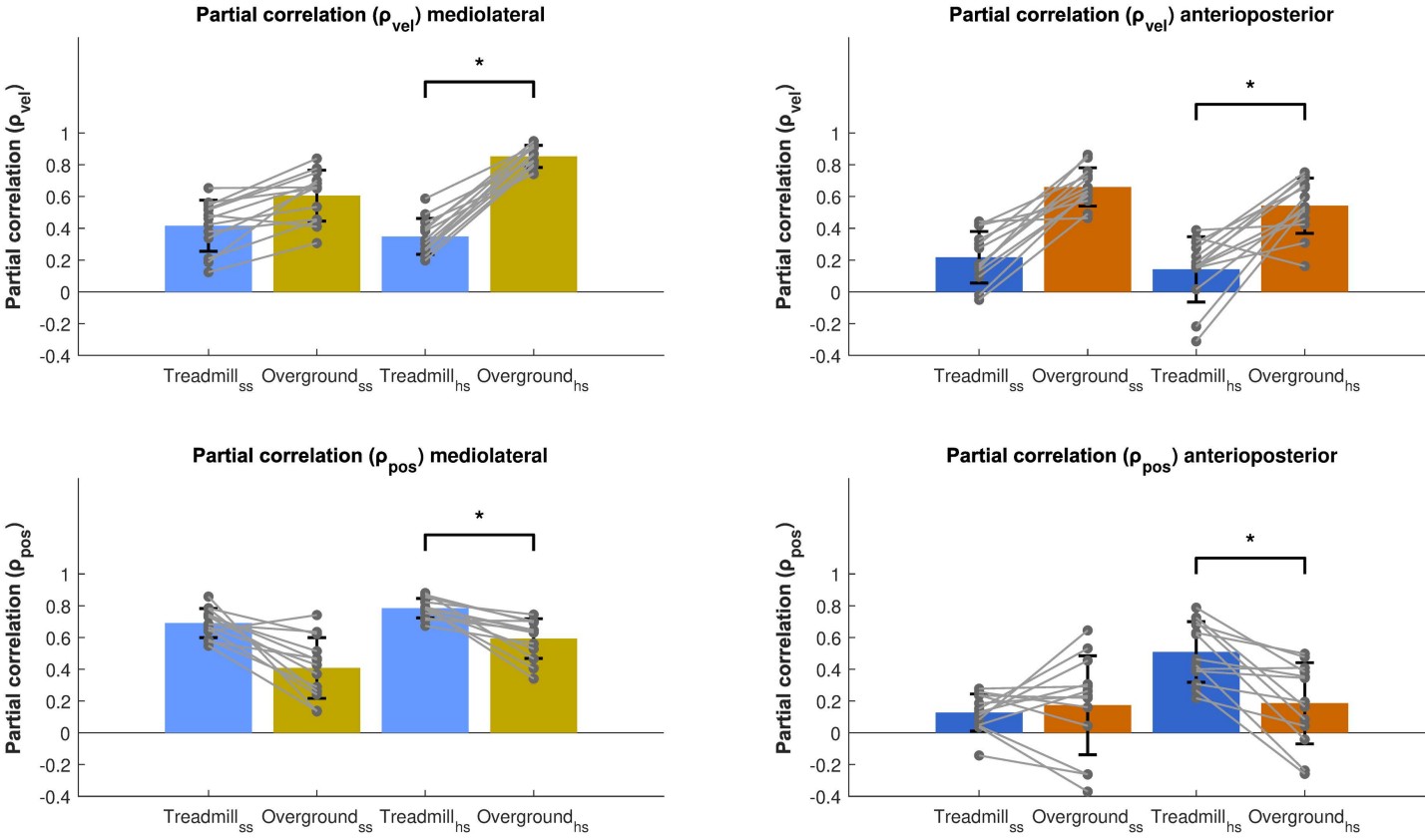

**Fig 6. Partial correlations of CoM$_{pos}$ and CoM$_{vel}$ with step width/length, comparing treadmill and overground walking.** The mean partial correlations have been plotted for the ML (left panels) and AP (right panels) foot placement models for overground and treadmill walking. Error bars represent the standard deviation and grey dots the individual data points. *$p < 0.05$.

## 4.1. Use of foot placement control during overground walking

Firstly, we hypothesized that during overground walking, we use feedback control [4] to adjust foot placement in response to deviations in CoM state, similar to treadmill walking (H1). Since the regression coefficients of the foot placement models (equations 1 & 2) were significantly different from zero (Fig 2), our results show that indeed CoM position and CoM velocity can predict our step widths and step lengths during overground walking. This finding supports H1, and is consistent with previous treadmill walking results [12]. For the ML foot placement mechanism, all participants showed only positive regression coefficients. A positive coefficient indicates that a more lateral (or medial) deviation in CoM position with respect to the stance leg is followed by a wider (or narrower) step. Similarly, in the AP direction, a positive regression coefficient indicates that a more forward (or backward) deviation in the CoM position is followed by a longer (or shorter) step. However, as can be seen from Fig 2, not all participants had positive regression coefficients in the AP model. A previous modelling study suggested that in order to meet the criteria for foot placement to achieve local stability, both the position and velocity gains need to be positive [25]. Thus, the negative regression coefficients observed in the AP model imply that we do not always prioritize stability in our selection of AP foot placement location. For example, prioritizing a change in speed may diminish stability related foot placement control in the AP direction. In such cases, we might instead rely on other stability control mechanisms that are not captured by foot placement control, such as ankle moment control

[5,20,26]. Nevertheless, at the group level, our results show that during overground walking, stability-related foot placement control is employed in both ML and AP directions.

We interpret the foot placement model to represent feedback control, with sensory information about the CoM state as input. This interpretation is supported by findings that sensory perturbations affect stability-related foot placement control and that swing leg muscle activity is associated with foot placement [10,12]. The CoM state is likely estimated based on multisensory integration, as proprioceptive [3,27], vestibular stimulation [28], as well as visual perturbations [14] have all been shown to alter foot placement responses.

From the start of the swing phase, the CoM state predicts foot placement better than the swing foot itself [9], and a gluteus medius burst during early swing predicts ML foot placement [10,12]. The foot placement model was used to quantify the precision and strength of this feedback control mechanism. Given that muscle activity during early swing phase has been associated with this mechanism, we used the CoM state at the start of the step (i = 1) as the predictor to gain insight into feedback control. To evaluate stability control performance, we used the model with the CoM state at the end of the step (i = 51) as the predictor.

Previous literature shows that the degree of foot placement control is modulated in response to stability demands and task constraints [19], indicating that foot placement control is context-dependent. Here, we investigated whether changing the context from treadmill to overground walking would affect the precision and strength of the foot placement responses (H2&H3).

## 4.2. Differences in precision of foot placement control between treadmill and overground walking

We hypothesized that foot placement precision would be higher during overground compared to treadmill walking (H2), given fewer stepping constraints during overground walking. Contrary to our expectation, the ML foot placement model showed the opposite pattern (Fig 4, *left panel*) with larger foot placement errors during overground walking. One possible explanation is that participants may perceive a reduced need for precise foot placement control during overground walking. Treadmill constraints may not only affect step-by-step foot placement control, but also average step width. Our results suggest that participants adopted a narrower step width on the treadmill than overground (Fig 3). From the literature, we know that when one walks with a wider average step width, step-by-step foot placement control can be loosened, while still maintaining comparable walking stability [19]. Thus, the narrower base of support on the treadmill might require more precise step-by-step foot placement in order to remain stable. Conversely, walking with wider steps during overground walking may reduce the need for tight foot placement control, resulting in higher foot placement errors. In addition, participants may perceive the stability demands of overground walking as less challenging, feeling more confident to loosen foot placement control, while adopting a more cautious strategy with increased foot placement precision during treadmill walking.

For the AP foot placement model no significant differences were found for foot placement precision. Thus, the tightness of step length control did not significantly differ between treadmill and overground walking.

## 4.3. Differences in the strength of foot placement responses between treadmill and overground walking

Our third hypothesis predicted stronger foot placement responses to variations in CoM state during overground compared to treadmill walking (H3). However, as shown and discussed above, foot placement control appears to be loosened during overground walking, which could make stronger responses more likely to destabilize gait. This was reflected in a lower position gain ($\beta_{pos}$) in overground as compared to treadmill walking (Fig 6), for the ML foot placement model both at the start of the step and at heel strike, and for the AP foot placement model at heel strike only. This pattern is consistent with a reduced urgency to attenuate variations in the CoM state through foot placement. However, the velocity gain ($\beta_{vel}$) demonstrated opposite effects. The velocity gain was larger during overground walking, indicating stronger responses to deviations in CoM velocity, in both ML and AP directions, both at the start of the step and at heel strike. Because these opposite

effects for $\beta_{pos}$ and $\beta_{vel}$ suggested a difference in the relative contributions of position and velocity feedback, we further explored this by computing and testing the partial correlations. The findings and their implications are discussed below under "Contribution of position and velocity feedback" and "Outlook" sections.

### 4.4. Gait parameters which affect step-by-step foot placement control

As mentioned earlier, average step width affects the degree of ML step-by-step foot placement control [19]. Apart from step width, this degree of foot placement control is also dependent on gait speed [23]. In this study, we did not control for gait speed experimentally, allowing us to investigate natural and comfortable walking behavior in both treadmill and overground conditions. Gait speed was higher and average step width wider during overground walking (Fig 3). In order to test the treadmill effect independent of the step width and speed effects, we controlled for the latter two in the statistical analysis. However, the differences in step width and speed are important to keep in mind when aiming to generalize tread-mill findings related to foot placement control to overground walking.

### 4.5. Contribution of position and velocity feedback

Since the position gain ($\beta_{pos}$) was smaller during overground walking compared to treadmill walking, while the velocity gain ($\beta_{vel}$) was larger, we conducted an exploratory analysis of the partial correlations of $CoM_{pos}$ and $CoM_{vel}$ with foot placement. At both the start of the step and heel strike, the partial correlation of $CoM_{pos}$ was lower during overground as compared to during treadmill walking, whereas the partial correlation of $CoM_{vel}$ was greater during overground as compared to during treadmill walking. As such, the relative contributions shifted between conditions, with position feedback contributing the most during treadmill walking and velocity feedback contributing more than position feedback during overground walking (Fig 8).

We identified step-by-step foot placement control during overground walking, underscoring that in both treadmill and overground walking conditions, foot placement is controlled with respect to the CoM state. However, the relative contributions of position and velocity feedback differ between the two. Visual information about walking direction and velocity is crucial for locomotion, arising from the relative motion between the eyes and the surrounding environment [29,30]. However, this optical flow is absent on the treadmill, and may create a mismatch between visual input and vestibular and proprioceptive feedback. Such sensory discrepancies may explain the greater reliance of spatial information from CoM movement during treadmill walking. On the other hand, while cadence, step length, and stride time can be adjusted to some extent on the treadmill, the preset speed may limit the ability to make individual adaptations, reducing the variability of CoM velocity [31]. The significantly lower variability of CoM velocity observed during treadmill walking may therefore limit the velocity-related feedback available for foot placement adjustments due to task constraints. In contrast, during overground walking, aligned sensory information from vision and body movement, together with fewer constraints on CoM velocity, may have led to a greater contribution of CoM velocity feedback to foot placement control. Our results support the hypothesis that foot placement control differs between treadmill and overground walking as can be seen by changes in CoM feedback contribution. Although an increase in variance could mathematically inflate the contribution of CoM velocity, the variance of CoM position was increased as well (S1 Fig 1 in S1 File). The former can therefore not fully explain the changes in partial correlations between conditions, making a sensory re-weighting more likely.

### 4.6. Outlook

Due to the observed differences in foot placement control, caution is warranted when translating findings from treadmill walking directly to overground walking or daily life gait. Our finding that the contribution of CoM velocity exceeds that of CoM position during overground walking is particularly relevant for training and rehabilitation interventions aimed at improving gait stability through enhancing foot placement control. To promote gait stability in daily life, interventions may therefore target the contribution of CoM velocity rather than position.

Notably, many existing training interventions, such as walking with force-field perturbations or muscle vibration, have mainly targeted and improved the position gain. A shift in focus towards training interventions that directly enhance the velocity gain may be warranted, particularly given that in stroke patients, the velocity gain appears to be the most disrupted [6]. In previous implementations of force-field perturbations and muscle vibration, the timing and magnitude of the applied stimuli were based on CoM position at the start of the step. Reprogramming these training devices to instead use the real-time CoM velocity at the start of the step as an input to adapt the perturbation and vibration intensity, may shift their impact from position gain toward velocity gain enhancements.

In addition, as feedback contributions differ between treadmill and overground walking, training exclusively on a treadmill may not yield the expected improvements in real-world gait stability. Nonetheless, treadmill training offers practical advantages, such as enabling longer, uninterrupted walking without the need for extensive space, making it a convenient tool for rehabilitation. Adding external perturbations or VR, that lead to changes in the steady-state stepping kinematics during normal treadmill walking, might be able to simulate real-world challenges more closely. Incorporating overground walking alongside treadmill-based rehabilitation may help facilitate transfer to real-world conditions and improve the overall effectiveness of the intervention.

The use of wearable sensors for real-world gait assessments is continuously increasing, yet accurately estimating spatial parameters, particularly the spatial relationships between different body segments, remains challenging [32]. Until now, CoM position was thought to be the primary factor in explaining foot placement control [23], making it difficult to assess with wearable sensors. However, our findings highlight the greater influence of CoM velocity during overground walking, which could serve as the foundation for developing methods to assess foot placement control based on acceleration and angular velocity signals from wearable sensors. While obtaining position information through double integration of the acceleration signal remains highly error-prone, our results indicate that the information of CoM position may not be strictly necessarily to estimate the quality of foot placement control. Instead, CoM velocity information with lower error through just a single integration, might be sufficient to interpret foot placement control as step length can already be estimated using IMU sensors, and initial approaches are developed for estimating step width [32,33]. This, in turn, could pave the way for evaluating foot placement control in daily life, which is a necessary step toward understanding the mechanisms underlying gait stability in real-world settings.

### 4.7. Limitations

The sample size of 13 participants is relatively small, but the number of steps included is well above the value for which foot placement outcomes tend to plateau [23]. Nevertheless, the generalizability of our results to a broader population may be limited, especially considering the high variability of results in the AP direction. Although we ensured that the overground walking steps were all straight ahead (we excluded the curved parts of the figure 8 paths), we cannot be certain that participants did not already adjust their steps in anticipation of the turn. To minimize such potential effects, we always excluded the last step of each straight walking path. Given the short walking distance, it was not desirable to remove more than one step at the end of each path. Therefore, some of the steps included might not reflect steady-state walking, potentially contributing to larger residual errors in overground walking. However, when repeating the analysis of foot placement errors during overground walking with an additional step removed before each turn, the result was unchanged compared to our results presented with just one step removed (S2 Fig 1 in S2 File). Another concern might be that, even after rotating the marker trajectories, a slight drift remains visible (Fig 1), which could potentially affect the results of the foot placement model. However, foot placement errors did not increase linearly over the trial (S3 Fig 1 in S3 File). While treadmill walking enforces a constant walking speed, walking speed during overground walking might vary between trials of a participant. However, as the participants were healthy young adults, fatigue-related effects on walking patterns were not expected, and intraindividual variations in walking speed were within 5% and therefore considered consistent.

Another limitation that should be noted is that although we interpret the foot placement model to represent sensory feedback control, with the foot placement errors representing either motor noise or looser control, like any model, (part of) the residuals might exist because the model does not fully capture the actual control parameters. The relatively high reported $R^2$ of the model [9], suggests that the model is a meaningful representation of step-by-step foot placement control, even in this simple linear form. This is further supported by the sensitivity of the model in distinguishing foot placement control between different populations [3] and following training interventions [15,16]. However, it cannot be dismissed that different or more complex models could potentially generate an even better representation. In its current form, the foot placement model generates residuals which can be interpreted to at least partly represent a foot placement error in the context of stability control. This is underscored by partial correction of this "foot placement error" in healthy adults by subsequent center-of-pressure shifts during single stance through ankle moment control. When external stabilization alleviates the demands on stability control, execution of this corrective mechanism is diminished [34], underpinning that the residual of the foot placement model has implications for stability control.

## 5. Conclusions

This is the first study showing that stability related foot placement control is a mechanism that is executed both during treadmill and overground walking. However, treadmill constraints alter foot placement strategies as well as CoM position and velocity feedback contributions compared to overground walking. During overground walking, we tend to walk with a wider average step width and looser step-by-step foot placement control. Moreover, the contribution of velocity feedback is greater in overground as compared to treadmill walking. These results call into question the degree to which treadmill findings on foot placement control translate to daily life gait, but they are promising for future home-based assessments.

## Supporting information

**S1 File. Variability of CoM position and velocity.**
(PDF)

**S2 File. Foot placement error with steps nearing the turn removed.**
(PDF)

**S3 File. Progression of foot placement errors over trial.**
(PDF)

## Acknowledgments

We would like to thank Lena Salzman, Zhongke Mei, David Aschwanden, Pietro Sartori, Stefania Bottoni, Livia Vinzens and Laura Fusi for providing us the data as well as Prof. Dr. William Taylor. Moreover, we are grateful to the StepuP consortium for allowing this collaboration.

## Author contributions

**Conceptualization:** Charlotte Lang, Jaap H. van Dieën, Anina Moira van Leeuwen.

**Formal analysis:** Charlotte Lang, Anina Moira van Leeuwen.

**Funding acquisition:** Deepak K. Ravi, Sjoerd M Bruijn, Jeffrey M. Hausdorff, Jaap H. van Dieën.

**Supervision:** Deepak K. Ravi, Sjoerd M. Bruijn, Jaap H. van Dieën.

**Visualization:** Charlotte Lang.

**Writing – original draft:** Charlotte Lang, Anina Moira van Leeuwen.

**Writing – review & editing:** Charlotte Lang, Deepak K. Ravi, Sjoerd M. Bruijn, Jeffrey M. Hausdorff, Jaap H. van Dieën, Anina Moira van Leeuwen.

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
