## [Decision Letter · Decision Letter 0]

16 Dec 2025

PONE-D-25-57410How do we tread? Differences in stability-related foot placement control between overground and treadmill walking in young adultsPLOS One?

Dear Dr. van Leeuwen,

We look forward to receiving your revised manuscript.

Kind regards,

Anne E. Martin

Academic Editor

PLOS One

“This research was financially supported by the EU Joint Programme – Neurodegenerative DiseaseResearch (JPND) to the StepuP consortium: Steps against the burden of Parkinson’s Disease,grant number JPND2022-128. Moreover, the experimenters responsible for the data collection were funded by the LOOP Zurich and the Vontobel Stiftung.”

“This research was financially supported by the EU Joint Programme – Neurodegenerative DiseaseResearch (JPND) to the StepuP consortium: Steps against the burden of Parkinson’s Disease,grant number JPND2022-128, obtained by the StepuP consortium (J.H. van Dieën, W. Maetzler, J. Hausdorff, M. Brodie, N. Singh & F. Laporta). Moreover, the experimenters responsible for the data collection were funded by the LOOP Zurich and the Vontobel Stiftung. The funders had no role in study design, data collection and analysis, decision to publish, or preparation of the manuscript.”

Reviewers' comments:

Reviewer's Responses to Questions

**Comments to the Author**

1. Is the manuscript technically sound, and do the data support the conclusions?

Reviewer #1: Yes

Reviewer #2: Yes

2. Has the statistical analysis been performed appropriately and rigorously?

Reviewer #1: Yes

Reviewer #2: Yes

3. Have the authors made all data underlying the findings in their manuscript fully available?

Reviewer #1: No

Reviewer #2: Yes

4. Is the manuscript presented in an intelligible fashion and written in standard English?

Reviewer #1: Yes

Reviewer #2: Yes

Reviewer #1: Manuscript Number: PONE-D-25-57410

Title: How do we tread? Differences in stability-related foot placement control between overground and treadmill walking in young adults

Review Summary

This study investigates the differences in foot placement control mechanisms between treadmill and overground walking in young healthy adults (N=13). Utilizing linear models that predict step width and length based on Center of Mass (CoM) state, the authors tested hypotheses regarding the existence, precision, and strength of these control mechanisms across conditions.

The results suggest that while foot placement control is active in both conditions, it manifests differently: overground walking is characterized by lower precision (larger residuals), wider steps, and a shift in reliance from CoM position to CoM velocity feedback. The authors conclude that caution is required when generalizing treadmill findings to real-world gait.

General Assessment

This is a well-written and logically structured manuscript that addresses a significant gap in the biomechanics literature: the ecological validity of treadmill-based stability research. The methodology regarding the foot placement model is grounded in established literature, and the statistical approach (using Linear Mixed Models to control for speed/step width confounds) is robust.

However, there are valid concerns regarding the specific experimental constraints of the overground condition (the Figure-8 path) and the sample size that need to be addressed to ensure the validity of the conclusions.

Major Comments

1. The Steady-State Assumption in a Figure-8 Path

The overground condition involved walking in a Figure-8 path, with analysis restricted to the straight segments. The authors removed the last step of the straight path to minimize anticipation effects. Though is is known that anticipatory locomotor adjustments for turning often begin several steps prior to the turn itself (e.g., changes in trunk roll, step width modulation). In a 10-meter straight section, excluding only the last step may not be sufficient to capture true "steady-state" walking comparable to the treadmill condition. Can the authors provide data or citations demonstrating that the steps analyzed (steps n-2, n-3, etc.) do not show preparatory kinematic changes (such as CoM deviation toward the turn direction). If this cannot be demonstrated ok,but i think this limitation should be discussed more prominently as a potential confounder for the increased "error" (residuals) found overground.

2. Interpretations of Velocity Gain βvel

The study reports a higher relative contribution of velocity feedback during overground walking. On a treadmill, the belt speed is fixed. While the subject can fluctuate relative to the belt, the global velocity is highly constrained. Overground, the subject has degrees of freedom to modulate speed freely. I believe that it is crucial to distinguish whether the increased βvel overground is a result of better sensory integration (as argued in the Discussion) or simply a mathematical result of there being more variance in velocity to regress against during overground walking. If the range of CoM velocity on the treadmill is negligible, the model may naturally suppress the βvel term.

Can the authos please provide descriptive statistics comparing the variance (or standard deviation) of the CoM velocity between the two conditions. If the variance is significantly different, the discussion should reflect that the "increased reliance" might be a function of the task constraints allowing velocity modulation, rather than purely a sensory re-weighting.

3. Sample Size and Power

The final analysis included 13 participants. While the authors state that the number of steps (n=130) is sufficient for the model to plateau, the statistical power to detect differences between conditions (treadmill vs. overground) relies on the number of participants. Given the high inter-subject variability often seen in gait strategies, N=13 is on the lower end for a study generalizing to a broad population. If a power analysis was conducted a priori, please report it. If not, acknowledge the small sample size as a limitation specifically regarding the generalizability of the LMM results, particularly for the AP direction where results were more variable.

4. Coordinate System Rotation and Drift

The authors note that marker trajectories were rotated to align with the walking path, but "a slight drift remains visible". In a linear regression model where CoM is a predictor, systematic drift could artificially inflate the residuals or skew the position gain

Please clarify if the data were de-trended beyond the simple rotation. If the drift correlates with the progression along the 10m path, it acts as a structured noise. A sensitivity analysis (e.g., checking if residuals increase linearly over the trial) would strengthen the validity of the "lower precision" finding.

Minor Comments

1. Explanation of i=1 vs i=51:

The distinction between "feedback control" (at i=1, start of swing) and "performance" (at i=51, foot placement) is mentioned. However, for readers less familiar with this specific modeling approach, it would be helpful to explicitly state why the start of the swing phase represents feedback planning, whereas the end represents the execution/outcome.

Recommendation

Major Revision.

The study offers valuable insights, but the interpretation of the results - specifically regarding the Figure-8 constraint and the mathematical influence of velocity variance – in my view needs to be more rigorously defended to support the conclusions.

Reviewer #2: The current manuscript tested the idea that step width and step length is modulated by the nervous system to ensure stability. Previous studies have primarily tested this idea on a treadmill while most walking by individuals is completed overground. Individuals walking overground and on a treadmill. A linear statistical model was created whereby step width was modeled as a function of center of mass position and velocity. Differences between the two walking conditions were detected specifically the authors report lower foot placement precision overground vs. treadmill with greater sensitivity in center of mass velocity compared to position. The topic is of interest to the general gait and biomechanics community. There are a few considerations with the manuscript in its current form that should be addressed.

1. Overall the introduction appears unnecessarily lengthy. While the information presented mostly relates to the purpose of the study, it could be tightened up for the reader. As an example paragraph 2 and 3 are very similar in topic and could be combined. Paragraph 4 and 5 are also similar and could be combined.

2. The primary critique is that this work suggests that the nervous system is targeting a particular step width on a step-by-step basis and if that step is not achieved it must be an error, despite the fact that many step placements could still satisfy the criteria for stability. Under conditions in which constraints are provided where a specific step width or range of step widths are fixed (e.g., Rosenblatt et al. Exp Brain 2014, Sidaway Exp Ger 2025) it is sensible that the constraints imposed would require greater precision because error in foot placement negatively effects task completion. (Sawers & Ting G&P 2015). However, overground walking unconstrained has no such constraints. People can and do meander. Thus, is it possible that the argument for unconstrained walking places too much emphasis on precision?

3. Introduction Line 47: The current study is not investigating falls or neurological diseases. The sample was 14 individuals with an average age of 24. Falls are less likely a concern. However, the manuscripts first paragraph describes falls and risk of falls. It is generally likely that whatever mechanism discovered here is likely not the same for people with high fall risk. Thus, the intro should focus on what we are learning about how the young nervous system controls stability. Consider moving patient population to the discussion.

4. While walking speed on a treadmill maintains a consistent speed, overground walking provides no external constraints to the nervous system. During overground trials individuals were walking straight then turning then walking straight on a 4-minute loop. Was speed consistent for overground trials and if not, could that modify model estimates?

5. The use of the linear model is well reasoned and justified to predict foot placement, however the interpretation seems to suggest that whatever variance is not accounted for has a biological interpretation, i.e., error in foot placement. This seems like leap. This model included two terms to account for position and velocity of the center of mass. It may also be that this simple model does not account for other factors that could influence step-to-step adjustment. Please add this as a potential limitation of the model.

6. Terminal swing is typically considered in textbooks such the Neuman textbook (Kinesiology of the Musculoskeletal System) is considered 87-100% of the gait cycle. In the current manuscript the authors seem to consider this a specific instant and time where in the model terminal swing i=51. Is this 1 frame back prior to foot strike? This should be clarified for the reader.

7. The discussion is very long and could benefit from tightening the language. Section 4.6 could be one or two paragraphs. As is it feels a bit bloated. Consider revising.

**Do you want your identity to be public for this peer review?** For information about this choice, including consent withdrawal, please see our Privacy Policy

Reviewer #1: **Yes:** Christos Chalitsios, PhD

Reviewer #2: No

---

## [Author Response · Author response to Decision Letter 1]

26 Jan 2026

The authors thank the reviewers for the attention dedicated to our manuscript. We feel like we have improved our manuscript based on your comments. A point by point response to your comments have been attached as a file.

---

## [Decision Letter · Decision Letter 1]

25 Feb 2026

How do we tread? Differences in stability-related foot placement control between overground and treadmill walking in young adults

PONE-D-25-57410R1

Dear Dr. van Leeuwen,

We’re pleased to inform you that your manuscript has been judged scientifically suitable for publication and will be formally accepted for publication once it meets all outstanding technical requirements.

Kind regards,

Anne E. Martin

Academic Editor

PLOS One

Additional Editor Comments (optional):

Reviewers' comments:

Reviewer's Responses to Questions

**Comments to the Author**

Reviewer #1: All comments have been addressed

Reviewer #2: All comments have been addressed

2. Is the manuscript technically sound, and do the data support the conclusions?

Reviewer #1: Yes

Reviewer #2: Yes

3. Has the statistical analysis been performed appropriately and rigorously?

Reviewer #1: Yes

Reviewer #2: Yes

4. Have the authors made all data underlying the findings in their manuscript fully available?

Reviewer #1: Yes

Reviewer #2: No

5. Is the manuscript presented in an intelligible fashion and written in standard English?

Reviewer #1: Yes

Reviewer #2: Yes

Reviewer #1: Thank you for addressing my comments. I have no further concerns regarding the manuscript. My recommendation is that the article can be accepted in the current form.

Reviewer #2: (No Response)

**Do you want your identity to be public for this peer review?** For information about this choice, including consent withdrawal, please see our Privacy Policy

Reviewer #1: No

Reviewer #2: No

---

## [Editor Report · Acceptance letter]

PONE-D-25-57410R1

PLOS One

Dear Dr. van Leeuwen,

I'm pleased to inform you that your manuscript has been deemed suitable for publication in PLOS One. Congratulations! Your manuscript is now being handed over to our production team.

Kind regards,

on behalf of

Dr. Anne E. Martin

Academic Editor

PLOS One